# Viewpoints of pregnant mothers and community health workers on antenatal care in Lweza village, Uganda

**Mackenzie E. Delzer** [1¤]*, **Anthony Kkonde**[2], **Ryan M. McAdams**[3]

**1** University of Wisconsin School of Medicine and Public Health, Madison, WI, United States of America,
**2** Mukono District Municipality, Mukono, Uganda, **3** Department of Pediatrics, University of Wisconsin School of Medicine and Public Health, Madison, WI, United States of America

¤ Current address: University of Wisconsin School of Medicine and Public Health, Madison, Wisconsin, United States of America
* mecarlson3@wisc.edu

## Abstract

### Background

Uganda is a low-income country with high fertility, adolescent birth, and maternal mortality rates. How Ugandan Ministry of Health antenatal education guidelines have been implemented into standardized health education and how pregnant women utilize health facilities remains unclear.

### Objective

We aimed to determine how women obtain education during pregnancy, what guidelines health educators follow, and what barriers exist to receiving antenatal care in Lweza Village, Uganda.

### Methods

Household surveys were conducted with women in Lweza who were or had previously been pregnant. Focus group discussions were conducted with community members and Lweza Primary School teachers. Interviews were conducted with key informants, including midwives, a traditional birth attendant, a community leader, and a Village Health Team member. Data collection was done in English along with a Luganda translator.

### Results

Of the 100 household surveys conducted, 86% of women did not meet the WHO recommendation of 8 antenatal appointments during their pregnancies. Reasons cited for inadequate visits included facing long wait times (>7 h) at health facilities, getting education from family or traditional healers, or being told to delay antenatal care until 6 months pregnant. Informant interviews revealed that no standardized antenatal education program exists. Respondents felt least educated on family planning and postpartum depression, despite 37% of them reporting symptoms consistent with postpartum depression. Education was also

**Data Availability Statement:** All relevant data are within the manuscript and its Supporting Information files.

**Funding:** Research support was provided by an award from the University of Wisconsin School of Medicine and Public Health and the Herman and Gwendolyn Shapiro Foundation. The funders had no role in study design, data collection and analysis, decision to publish, or preparation of the manuscript.

**Competing interests:** The authors have declared that no competing interests exist.

lacking on the use of traditional herbs, although most women (60%) reported using them during pregnancy.

## Conclusions

Most women in Lweza do not receive 8 antenatal appointments during their pregnancies or any standardized antenatal education. Educational opportunities on family planning, post-partum depression, and the safety of traditional herbs during pregnancy exist. Future studies should focus on ways to overcome barriers to antenatal care, which could include implementing community-based education programs to improve health outcomes for women in Lweza Village.

## Introduction

Worldwide, maternal mortality remains excessive, with almost all deaths occurring in low-resource settings; most of these maternal deaths are preventable [1]. In 2017, 66% of the estimated global maternal deaths happened in Sub-Saharan Africa, a region that also continues to have the highest global under-5 mortality (76 deaths per 1,000 live births in 2017), with almost 1 million neonatal deaths occurring annually from 1990 to 2017 (0% decline) [2]. While significant progress has been made in the last decade to improve health outcomes in Sub-Saharan Africa, the current state of infant and maternal health in countries like Uganda requires further advances. On average, each woman in Uganda will give birth to 6 children, which is more than the average for Sub-Saharan Africa and well over two times the fertility rate of the United States [3]. Unfortunately, in low income countries, like Uganda, limited access to healthcare facilities and resources, poor sanitation, and infectious diseases pose a significant childbirth risk to both maternal and fetal health. In Uganda, the maternal mortality rate is 375 per 100,000 women, whereas the rate in the United States is only 14 per 100,000 [4]. Childbirth in Uganda is also complicated by high rates of teen pregnancy and adolescent births. While the United States has an adolescent birth rate of 21.2 per 1,000, the Ugandan adolescent birth rate is nearly five times higher at 118.8 per 1,000 [4]. In addition to the high adolescent birth rate, recent studies have shown that adolescent girls are in the most need for education and access to maternal health services, including antenatal care [5]. Given the myriad of risk factors that threaten the maternal and infant health in the perinatal period, it is imperative that Ugandan women receive adequate antenatal care.

In Uganda, women who seek antenatal care and childbirth services have access to a continuum of care options within the healthcare system. The most basic level of care is provided by Village Health Team members, who exist in each village and often act as messengers for government health initiatives or local clinic events. The next tiers of care include Health Centers I, II, and III that provide more services and are staffed by providers with more advanced skills and knowledge. Pregnant women are able to obtain antenatal care beginning at the Health Center III, but labor and delivery services are not available until they reach the District Health Center IV where the vast majority of women attend their antenatal care appointments and present for childbirth. Women with obstetric emergencies or suspected complex childbirths need referrals to Health Center IVs, General hospitals, or National Referral Hospitals. Most services provided at these government health facilities are free, whereas private hospital clinics charge fees for their services. In Uganda, antenatal care and delivery services in rural and urban settings often involve traditional healers and traditional birth attendants (TBAs). TBAs

are typically trained by previous generations of healers with knowledge of ailments and treatments being passed down from their mothers or family members. It is common for practices to differ between healers from different families or in different locations.

The World Health Organization (WHO) has been invested in improving maternal and child health worldwide. WHO provides recommendations for antenatal care to address health disparities that exist in developing countries. WHO recommends that women attend at least 8 antenatal care appointments during each pregnancy to reduce the likelihood of stillbirth, to identify and manage potential pregnancy complications, and to provide health education at multiple visits [6]. Antenatal care for women is essential to update immunizations; to educate about infection-prevention, nutrition and tips for a healthy pregnancy; to screen for health problems that would indicate high-risk pregnancy; and to familiarize women with the health facilities [7]. In Uganda, pregnant women often do not attend regular antenatal care appointments and only utilize maternal health options when they feel ill [8]. In addition, only a minority of women initiate antenatal care early in pregnancy, a delay in care that limits the amount of contact a woman has throughout her pregnancy, which may have a negative health impact in the perinatal period [9]. WHO recommendations for community mobilization to improve maternal and newborn health, especially in areas with limited access to antenatal care, include participatory learning and action through women's groups [10]. According to a 2017 study, when these WHO guidelines are utilized in rural settings with low access to health services, there is an associated improvement in home delivery practices that contribute to a reduction in neonatal mortality [11]. This study aimed to gain a better understanding how Ugandan women living in a rural village setting obtain education during pregnancy, what guidelines health educators follow, and what barriers women face in obtaining antenatal care. Insight into current antenatal care practices, pregnancy preparation in communities, and viewpoints of pregnant mothers and community health workers will help in the future development of recommendations to enhance antenatal care delivery methods and utilization to improve maternal and child health.

## Materials and methods

### Ethical statement

Ethical approval was received via the University of Wisconsin School of Medicine and Public Health Institutional Review Board (IRB). The Principal Medical Officer of Mukono wrote a letter to the IRB committee expressing support for the project and stating that IRB approval was not necessary for the scope of the project. The chairman of Lweza Village signed a letter of approval for participation in the study. Written consent was given by the Principal Health Officer of Mukono Municipality, Uganda to conduct the study in Lweza. Verbal consent was given by all participating community members and leaders; declining to be involved in the study did not affect any woman's treatment. Verbal, rather than written, consent was obtained as approved by the IRB. Given that the study was occurring in an area of low literacy and was being conducted verbally, verbal consent was deemed appropriate. A consent statement was read aloud and participants signed or used their thumb print to agree to the verbal consent on the questionnaire. No identifiable patient information was included in the questionnaire. Women under the age of 18 years were excluded.

### Study design and analysis

An exploratory, community-based, cross-sectional study was conducted, in the form of a semi-structured patient questionnaire with open and closed question types. Both quantitative and qualitative approaches were taken with the understanding that the qualitative data was

necessary for illustrating community member experiences with pregnancy and health care. A variety of antenatal care topics were discussed with community women who access these services, midwives, community leaders, and a Village Health Team member who provides care and education. Data collection tools included a standardized questionnaire, in-depth interview question guides, and focus group discussion guides. Each of these guides explored various aspects of antenatal care including: number of pregnancies, antenatal care practices, health education topics, nutrition and medications, breastfeeding, family planning, HIV and malaria transmission, postpartum depression, and preparedness for the first baby. As a part of the household questionnaire, women were asked to rate their knowledge of various health topics on a scale of 0 to 5, where 0 represented that they knew nothing about the topic and 5 represented they knew everything about that topic, even enough to teach someone else.

To assess education during pregnancy and barriers to receiving antenatal care in rural Uganda the authors (M.D., A.K., and R.M.) developed a questionnaire. The questionnaire was pilot tested with two women from Lweza to ensure proper wording and adequate translation of the questions, but was not validated. The household interviews were done in the participants language of choice and were led by the translator who conducted the questionnaire in a verbal manner and filled out the appropriate responses on the questionnaire. The translator was not part of the research team. The in-depth interviews and focus group discussions, conducted mostly in English and occasionally in Luganda, were translated to English in real-time as needed. The key informant interviews and focus group discussions were all recorded with permission and later transcribed.

The questionnaire, which involved 47 questions, most of which (45/47) were closed-ended, included questions about patient demographics, pregnancy history and information, health education and practices, and rating the level of knowledge about various pregnancy health topics. Closed-ended question data were entered in Excel and descriptive statistics were used to analyze the results. Quantitative data were analyzed using IBM SPSS Statistics for Windows, build 1.0.0.1508 (IBM Corp., Armonk, N.Y., USA) for descriptive statistics such as frequencies and percentages.

Open-ended question responses were manually analyzed by the authors using a thematic approach in which similar responses were grouped into thematic categories. To limit author bias, no responses were eliminated. Direct respondent statements were included to avoid translation bias. Qualitative data (e.g., open-ended question responses) were transcribed and analyzed by the authors using thematic analysis following the steps similar to those recommended by Braun and Clarke [12]. To become familiar with the data, the first author (M.D.) read through all the transcripts and field notes recorded from the standardized questionnaire, in-depth interviews, and focus group discussions multiple times. Data were coded, manually analyzed, and grouped by assessing for relationships between them to form sub-themes. Based on this process, the key themes examined included navigating the available healthcare systems, pregnancy health education, herbal medicine, common community concerns, family dynamics, and family planning.

## Study area

This study was performed in Lweza Village, Uganda, which is located in the Mukono District, just east of Kampala, Uganda's capital city. Mukono is comprised of 602 villages and spans nearly 3,000 square kilometers. Most of the population belong to the Baganda ethnic group, the largest kingdom in Uganda's central region.

The major public health care facility in Mukono is the Mukono District Health Center IV, a government hospital that provides numerous services including primary care, laboratory

testing, surgery, mental health, dental and eye care, pharmacy, radiology, HIV/AIDS, TB, and maternal and child health (antenatal, maternity, family planning, post-natal, immunizations). Each year, this 45 bed facility serves thousands of patients, including performing 7,789 deliveries, 15,126 antenatal care visits, and 1,708 cesarean sections [13]. The Mukono District Health Center IV is located in close proximity to Lweza Village, only about a 3 km walk or boda boda (motorcycle taxi) ride away. In addition to this health facility, private hospitals are available to patients throughout the district. Besides these formal health care services, traditional medicine and herbal medicine facilities are available, including TBAs for pregnant and laboring women. Each village elects two Village Health Team members who are responsible for community health promotion, health education, and passing along health messages from the government hospitals.

## Study population and sampling

The study target population was women who currently reside in Lweza Village and who had given birth in the Mukono District. Lweza Village was selected due to prior established relationships between the research team and the community. Lweza Village has a population of about 7,000 people. The research team, comprised of a University of Wisconsin School of Medicine and Public Health medical student (M.D.), a translator, and a Village Health Team member, started from a central location in Lweza and randomly selected a direction to begin household interviews. The translator and Village Health Team members were not part of the research team. The Village Health Team member identified households with eligible women for interviewing and every other household was interviewed until the team reached the end of the village. A new direction was then randomly selected and the interviews were carried out in this new direction working back through Lweza Village until there were 100 respondents.

In addition to the household interviews, six key informant interviews were held. Informant participants were selected based off of recommendations from the Mukono Municipality Principal Medical Officer. After permission was obtained, in-depth interviews occurred in private spaces at each informants' place of employment. The informants were compensated for their time with 10,000 UGX or roughly 2.71 USD. The key informants included: two government midwives, one private midwife, a local TBA, a Village Health Team member from the neighboring village of Basiima-Kikooza to minimize bias, and a leader from the Child Care and Youth Empowerment Foundation (CCAYEF). The CCAYEF is a local organization that was established to provide a gathering place for young girls and teenage mothers to share experiences, learn skills, and provide support to one another.

Two focus group discussions were conducted. These group discussions were arranged by a local translator who was familiar with the community. The first focus group discussion was with ten teachers at the Lweza Primary School. The second focus group discussion was with nine women who were all part of the Village Health Project Uganda, a local community organization that meets weekly for learning activities and participates in local sustainable projects like sack gardening or building rain-water tanks. Each of these focus group participants were compensated a small amount of money for their time.

## Results

### Social and demographic characteristics of the respondents

A total of 125 women participated in this study: 100 by household interviews, 6 by key-informant interviews, and 19 by focus-group discussions. All household-interview and focus-group-discussion participants were Lweza Village residents. Key-informant interviewees included Lweza residents and Lweza health facility employees. Of the 100 gathered responses,

all met the inclusion criteria and were analyzed. The demographics of the cohort are shown in Table 1. All women ($n = 100$) answered in the language of their choice, either English or Luganda.

Most (92%) women interviewed for the household surveys had less than a tertiary education. The American equivalent of tertiary education would be a technical school certification. Most women stated some type of religious affiliation. Of the household interviewees, most (69%) women had had three or more pregnancies with over half of them (54%) having their first pregnancy as a teenager. Most (86%) women did not meet the WHO minimum criteria of 8 antenatal care appointments during their pregnancies.

The literacy level of women interviewed was not obtained, however a general idea of literacy level may be inferred by highest level of education which was obtained. The education sessions conducted at the government health facility in the morning were primarily conducted in a verbal manner. It is uncommon to give women reading materials at their appointments; however, at the antenatal clinic, there are a few posters displayed with some educational materials.

## Common antenatal and pregnancy concerns

The common themes that were discussed in the household interviews, focus group discussions, and key informant interviews are summarized in Table 2.

**Table 1. Demographic and pregnancy characteristics of household interviewees (n = 100).**

| Age | n |
|---|---|
| 19 | 2 |
| 20–24 | 22 |
| 25 and older | 76 |
| **Highest level of education** | |
| None | 5 |
| Primary | 35 |
| Secondary | 52 |
| Tertiary | 6 |
| Degree | 2 |
| **Religious affiliation** | |
| Catholic | 31 |
| Protestant (Church of Uganda) | 21 |
| Pentecostal | 22 |
| Muslim | 21 |
| Seventh Day Adventist | 5 |
| **Number of pregnancies** | |
| One | 12 |
| Two | 19 |
| Three | 23 |
| Four | 18 |
| Five or more | 28 |
| **Age at first pregnancy** | |
| 19 or under | 54 |
| 20–24 | 37 |
| 25 or older | 9 |
| **Number of antenatal appointments during first pregnancy** | |
| Less than four | 28 |
| Four to seven | 58 |
| Eight or more | 14 |

**Table 2. Common interview themes.**

| |
|---|
| Going to an antenatal care appointment takes anywhere from a few hours to an entire day, with health education talks only happening in the morning. *(Villagers, Focus Group Discussions (FGDs), Key informants)* |
| Women receive pregnancy health information from each other while waiting at antenatal clinic, conversing with neighbors, and often from mothers-in-law. *(Villagers, FGDs)* |
| Local herbs are an integral part of pregnancy health, but these herbs sometimes complicate the medical care. *(Villagers, FGDs, Key informants)* |
| Most women and care providers worry about transportation for laboring women at night and/or to a referral hospital because of high cost. *(Villagers, Key informants)* |
| Community members and care providers recognize that teen pregnancy is a problem and often attribute this high incidence to poverty, young girls dropping out of school, and lack of parental responsibility. *(Villagers, FGDs, Key informants)* |
| While husbands often play a critical role in financially supporting a woman's pregnancy, they can also be a significant source of distress for women. *(Villagers, FGDs)* |
| Women want to know more about family planning, but the topic is complicated by the social norms surrounding the topic. *(Villagers, Key informants)* |
| To increase access to pregnancy health education, care providers would like to see more community outreach and education efforts in the villages themselves rather than at health facilities. *(Key informants)* |

The household questionnaire (Fig 1), in which women rated their level of knowledge on various pregnancy health topics, demonstrated the least amount of health education on the following four topics: postpartum depression (3.38/5), obstetric complications (3.2/5), HIV transmission (3.14/5), and family planning (3/5). The household questionnaire also asked women to report if they have ever experienced symptoms of postpartum depression. Two women declined to answer, and 36 out of the 98 respondents (37%) reported having experienced symptoms of postpartum depression.

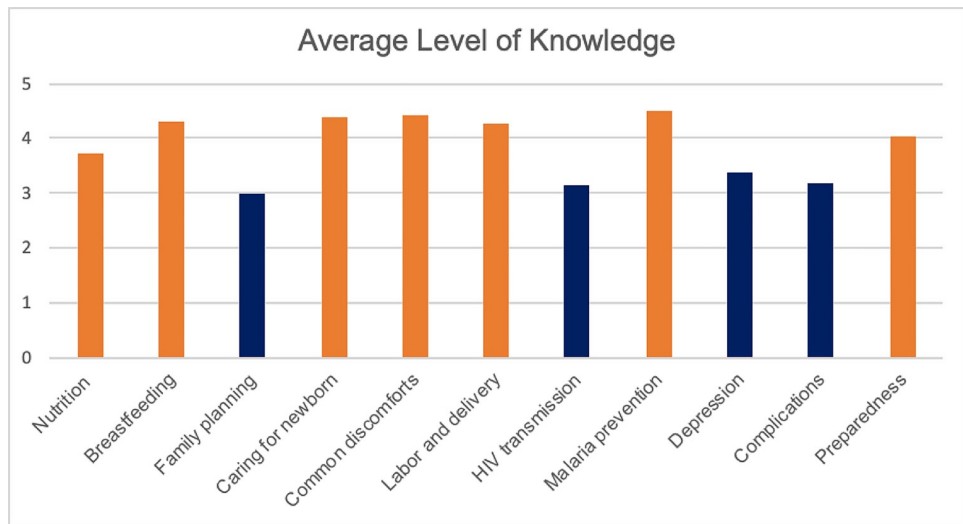

**Fig 1. Average level of knowledge.** This figure shows the averages of how women (n = 100) rated their level of knowledge on various pregnancy health topics on a scale of 0 to 5. Zero indicates that they know nothing about the topic, and 5 indicates that they know everything about the topic, even enough to teach someone else. The dark blue bars indicate the four lowest-rated health education topics: postpartum depression (3.38/5), obstetric complications (3.2/5), HIV transmission (3.14/5), and family planning (3/5). All of the other health topics were rated at a 3.74/5 and above.

### Sources of antenatal health education

Part of the household questionnaire asked women to report where they received their antenatal health education and what they could recall learning about (Fig 2). Most women (77%) reported getting their education from the government health facility. Less common were reports of getting antenatal health education from private health facilities (38%), relatives (29%), neighbors (12%), and TBAs (9%). The educational topics that the women reported spanned from antenatal care practices, including nutrition and hygiene, to techniques for caring for a newborn like bathing and breastfeeding. Many women recalled learning about nutrition, but when asked about diet change later in the survey, 57% of women changed their diets based on self-determination rather than based on advice from a midwife or health worker.

Based on observation, breastfeeding in the village seemed to be universally promoted by women, midwives, Village Health Team members, and TBAs. There is a cultural emphasis on breastfeeding, and access to formula is limited. In cases where breastmilk supplementation is

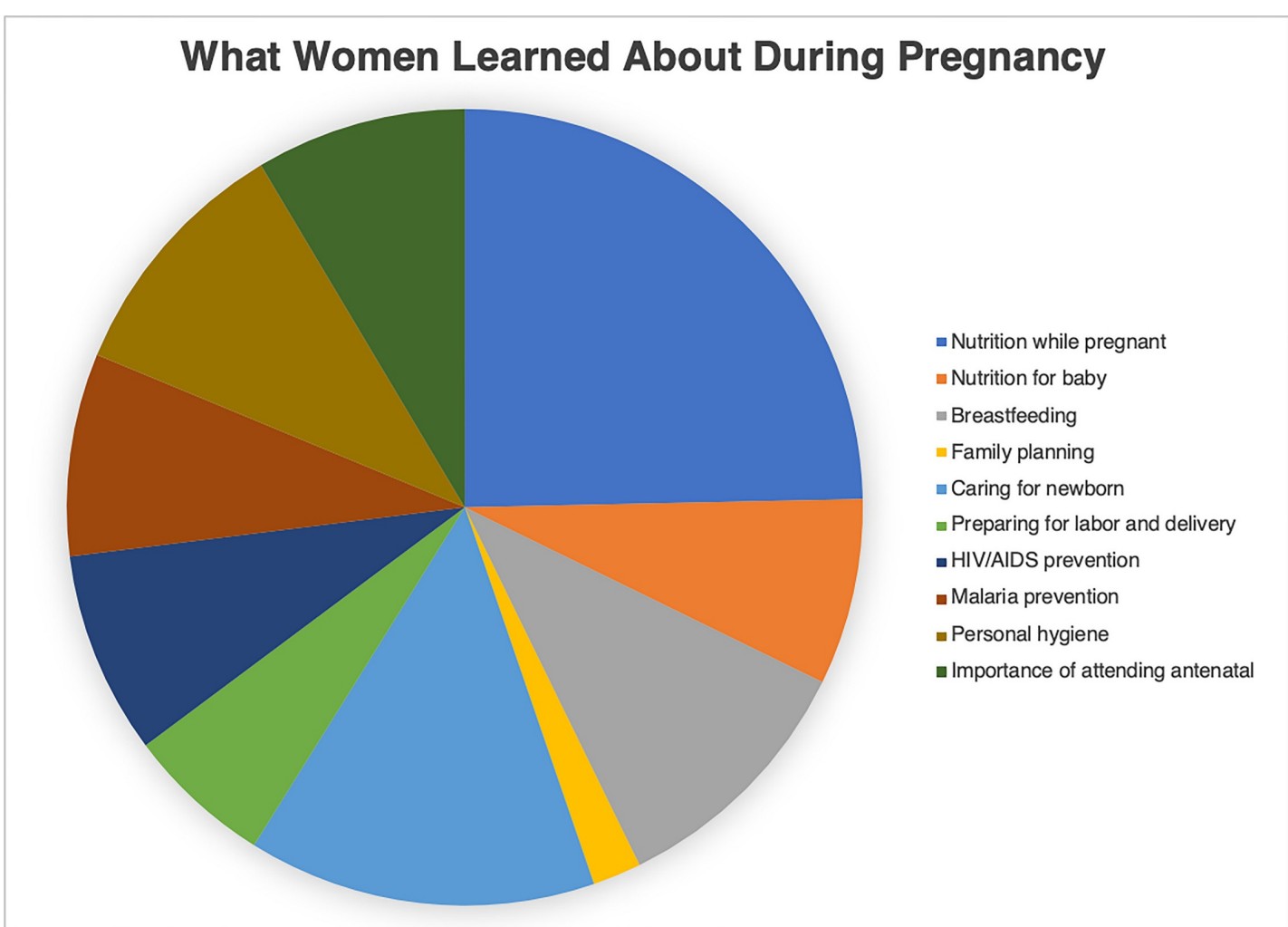

**Fig 2. What women learned about during pregnancy.** This figure shows the percentage of women surveyed (n = 100) that recalled learning about a variety of pregnancy health education topics. The topics recalled the most include nutrition while pregnant (75%), caring for a newborn (43%), personal hygiene (32%), and breastfeeding (32%). The topics recalled the least include family planning (6%), preparing for labor and delivery (18%), post-partum depression (0%), and obstetric complications (0%).

warranted, like in HIV-positive mothers or where production of breastmilk is lacking, it is commonplace to encourage supplementation with cow's milk, which may be easier to access and more affordable than formula.

Most women had experience with antenatal care at a government health facility. Most of the services, including routine antenatal care, at Mukono Government Health Facility IV were free. For these reasons, many women chose to attend at least one antenatal care appointment to get their pregnancy card, which held antenatal information and was required to be turned in when a woman showed up in labor in order to ensure timely and fair service. Most women reported that attending an antenatal care appointment was a long process that required patience. Women were told to be in line to register for their antenatal care appointment by 8:00 a.m. after which they were seated in a waiting area where a midwife, a HIV/AIDS counselor, or a nutritionist selling products gave an approximate 20-minute health education talk. Following this educational talk, women would typically wait up to 6 hours to be seen by a midwife.

As the Lweza Primary School teachers were explaining the process, they complained about the long wait times:

*You may (be finally called for your appointment and are feeling). . . dizzy because you are very hungry. . . you spend your whole day waiting to be checked (not having eaten anything). You can even collapse or be dizzy*! [Participant, FGD VHP-Uganda]

Women reported that certain people would also get special treatment and would skip the lines, making everyone else's wait time longer. If you were related to a health worker or had some money to pay, you might be able to skip the line. One woman reported,

*If you go with your husband, of course you jump the queue. Because they see that the husband is going back to work, so they don't want to waste his time.* [Participant, FGD Lweza Primary School]

One of the benefits of the long wait times, however, is that it allowed women the opportunity to bond with other pregnant women. As participants explained in the focus group discussions:

*They can tell you come very early, by 8 you are already there. But they may begin their service late. So as you sit there you are not just going to sit. We converse*! [Participant, FGD Lweza Primary School]

*You converse and get experience from the people who have already given birth.* [Participant, FGD Lweza Primary School]

Besides sharing stories, women might also use the time in line to purchase or collect items, which they assemble together into a "mama kit" which they are supposed to bring with them when they are in labor. This mama kit contains essential items women need to carry in case they deliver outside of a health facility (e.g., on the side of the road). Women who miss the health talk of the day had to wait until another appointment to get the information. Because of the rotation of topics, women might hear the same health talk twice and miss other topics completely:

*Once you find the talk is already done, it is done. You've missed that one, the next time you are reminded to come early.* [Midwife, Key informant interview, Government Health Facility]

## Balance between government health regimens and traditional medicine regimens

Many women reported that they sought advice or herbal medicines from one of the local TBAs. Some reasons cited for taking herbs included:

*Rejuvenate strength, most women become weak when pregnant, they take herbs mixed with clay. . . drink it and helps with baby too- just like an injection. [Participant, FGD VHP-Uganda]*

*Mix with water, you bathe. . . helps when she goes to deliver she can just push, so that the baby can be peaceful. [Participant, FGD VHP-Uganda]*

Most often, women were interested in obtaining emumbwa, a combination of clay from the soil and traditional herbs. Emumbwa seemed to be the most talked about herb and helped with:

*Giving energy, giving blood. . . soften the bones. . . helps those ones who have much more warmth in the stomach which can be bad for the baby, bring out the baby. . . treating syphilis, gonorrhea, candida. . . early morning sickness. . . bring appetite also. . . and energy. [Participants, FGD Lweza Primary School]*

If women were not obtaining these herbs from TBAs, they would get them from the bush themselves, from mothers-in-law, community members, or other experienced mothers who knew about collecting and preparing herbs. When interviewing the midwives at the government health facility, they expressed concern over the use of traditional herbal medicine. When a midwife was asked if the midwives recommend herbs, she responded:

*Not at all! We are actually moving away from, we call it native medicine, though Africans, we are so much attached to our culture. . . Here when you're pregnant, there's an assumption that if you don't use herbs you won't be able to produce normally. . . We encourage them to be active, and eat in time, and eat good food, but the herbs is still a challenge. . . What we tell them is it's not bad but we don't know how much. . . if you are taking a herb, a full cup of an herb, you may not know how much is contained in the what? In the herb. [Government Midwife, Key informant interview]*

This midwife mentioned that the government health facility routinely provides deworming agents like mebendazole because the herbs that pregnant women take are prepared with clay from the earth, which places women at risk for acquiring worms.

A TBA expressed that there needs to be better collaboration between the people in her field and the health workers at the hospitals. She explained the importance of receiving antenatal care at the government health facility and described her role as one to supply herbal and traditional meds throughout a woman's pregnancy as well as help with delivery. She also refers women to the government health facility when there are complications. She described the process of when a woman hemorrhages as follows,

*When she overbleeds, and is feeling dizzy, I get a bottle of mirinda fruit [a local berry-flavored soda], mix it with boiled eggs and with tomato. . . and make a syrup, which I give to the mother. . . then rush her to hospital. [Traditional birth attendant, Key informant interview]*

## Teenage pregnancy as a community health concern

Throughout the key informant interviews, respondents were prompted to offer suggestions for improving the health of their communities. Overwhelmingly, the responses were about prevention of teen pregnancy. During an interview with a coordinator at the CCAYEF, an organization created to educate and empower girls and teenage mothers, the coordinator shared thoughts regarding the consequences of teenage pregnancy:

> *It's a problem because it puts those girls at risk when they are giving birth. It's a problem still because most of the men who impregnate those young girls run away and don't take responsibility. . . It's a problem because young girls drop out of school. . . This girl has no job. . . Teenage pregnancy prevents the girls from growing up normally. . . denies them a childhood. Lack of support from parents to receive the girl back as a child [after giving birth] means she thinks "I should get married because I'm now big". . . The laws should start to work, sex education to girls, empowering the parents and even teachers in school [are suggestions for changing this]. [CCAYEF Coordinator, Key informant interview]*

These sentiments were echoed in each of the midwife interviews, in both focus group discussions, and in the household interview conversations with young mothers who had many children.

## Postpartum depression

The results of the household interviews showed that 36% of women reported feeling symptoms of postpartum depression. Only half of those women who were feeling depressed sought help for their symptoms, and usually they relied on talking to a family member. Despite a high postpartum depression rate, none of the respondents recalled learning about depression when asked to recall pregnancy health education topics earlier in the survey.

When asked for reasons for the depression, most women reported being stressed about financially supporting her baby. They struggled with inadequate food, clothing, and proper sanitation. Some women reported being thankful for the help of their husbands while others said that their husbands contributed to their anxiety. Additionally, many women reported that having a baby, which was the opposite sex than they had wanted was a significant source of distress. Women also mentioned that going through complications with pregnancy and delivery left them feeling depressed afterwards.

## Discussion

Limited literature exists describing the views and opinions of Ugandan women from rural areas regarding antenatal care education and provision. The goals of this study were to delineate how antenatal care education is disseminated, how antenatal care is obtained, and what local practices affected that care. Based on our study, the current model of pregnancy health education delivery at the government antenatal care appointments in the Mukono District leaves a large proportion of women with inadequate health information during their pregnancies. Previous studies have shown that women equate higher quality of care when they feel adequately informed about their pregnancy, the process of labor, and complications that may arise [14]. Insufficient antenatal education may increase the risk for poor pregnancy outcomes if women are unable to attend their appointments and fail to receive important health information. This study identified antenatal care-related topics where educational gaps exist, which can help guide future efforts to enhance antenatal care education. An example of this is

incorporating education on postpartum depression, an area where antenatal care education is lacking despite the high rate (36%) of women reporting postpartum depression symptoms.

This study identified areas of opportunity for improving education, access, and practices surrounding antenatal care. A need exists for standardized pregnancy health education topics and finding creative ways to get that information to communities that inconsistently utilize government antenatal care services. A current strategy to disseminate antenatal health education throughout the Lweza population involves midwives at the Mukono District Health Center IV who are attempting a new system to streamline the education women need. In this new system, women are split up into groups by age and gestational age (e.g., teen mothers are put together, women in their second trimester are put together). Health talks are then tailored to the specific groups so women are able to bond with each other and share experiences more easily. This promising approach requires further study to determine if it will yield improved antenatal care educational outcomes.

Several barriers were identified that restrict the amount and type of pregnancy health education women receive in Lweza Village. First, the high teen pregnancy rates likely complicate health education, since teens are at risk for increased pregnancy-related health complications, are more likely to die in childbirth, and are also often socially disadvantaged [5]. Second, the current 30 minute morning education session model at the government health facility seems ineffective at reaching the pregnant population. With only 14% of this study population meeting the WHO recommendation of 8 antenatal care appointments, better ways to engage and educate women during pregnancy are needed. Another health education barrier includes the various, and sometimes conflicting, sources of antenatal health education. This barrier was expressed by community members, the herbal medicine provider, and the midwives. A unified health education delivery approach, with collaboration between the various antenatal care entities, may allow a more comprehensive care mode; however, this approach would be challenging given the current discrepant messages women receive, ranging from WHO recommendations to advice from traditional herbalists.

The focus group discussions also highlighted opportunities for improvement in the process of obtaining high-quality antenatal care, all of which might encourage further use of antenatal care services in this community. Increasing positive experiences at the antenatal care health facilities has the potential to encourage continued and more frequent use of antenatal health services. An opportunity to improve the perceived quality of care includes strengthening the relationships between midwives and pregnant women [15]. Increased use of antenatal care services during a pregnancy has been shown to increase the likelihood of women delivering in a health facility where they will have skilled birth attendance, creating the potential to prevent a significant majority of maternal deaths [7, 14].

Our study has limitations, including the sampled population and tools for postpartum depression screening. Due to IRB restrictions regarding exclusion of minors, we were unable to sample teenagers, who represent the majority of the population who had recently given birth. Because of this, many women who were interviewed were older and some had not been pregnant in several years. Our questionnaire was unvalidated for the local culture, which may have impacted participant responses and interpretation of their responses. Some of the data obtained from the household surveys may not represent the perspectives on the most current health practices. Additionally, translating the household survey questions regarding postpartum depression was a challenge because the understanding of depression as an illness was limited, so in some cases these translations may not reflect the intended response. For example, depression, as opposed to disappointment or sadness, was a hard term to describe to women and the questions regarding this were limited. Depression was assessed using a recall question rather than assessing depression using a validated screening tool (e.g., Patient Depression Questionnaire-9). However, a significant number of women reported symptoms of depression,

so a more standardized screening process would be helpful in future studies to accurately determine the women who had postpartum depression.

This study revealed several areas of opportunity for future work in this community. Given the impact that postpartum depression has on healthy cognitive and socio-emotional development of children [16], a follow-up study in the Lweza community to more formally assess for postpartum depression using standardized, validated tools would be beneficial. Other impacts of postpartum depression include a lower likelihood to breastfeed, greater adverse childhood outcomes, less of a likelihood to vaccinate the child, marital discord, and thoughts of infanticide [16, 17]. Training Village Health Team members who live in the communities and interact with new mothers on how to use a depression screening tool may provide a way to better assess the incidence of postpartum depression. Interventions tested in low- and middle- income countries show that training community health workers on psychosocial and psychotherapeutic techniques of interpersonal psychotherapy and cognitive behavioral therapy allows for culturally appropriate treatment of mild-to-moderate perinatal depression in rural settings where access to trained mental health providers is limited [18]. If this model was employed by training Village Health Team members on therapeutic techniques to use with depressed individuals, there could be a large impact on the health of mothers and the wellbeing of their children in Lweza.

Most women reported using herbal medicines during pregnancy, yet little is known about the safety of these medicines on the mother and fetus. Emumbwa, a supplement commonly used during pregnancy, is made from soil and clay from the ground combined with a mixture of various local herbs. Consumption of this mixture could be considered a form of pica. Pica is the craving and consumption of nonnutritive substances, including soil, which is a global practice [19]. Pica is prevalent in Sub-Saharan Africa, especially in pregnant women, and while the exact etiology is unknown, pica may help satisfy micronutrient deficiencies and is considered a cultural norm [19, 20]. Future investigations could determine how herbal mixtures are prepared, paying special attention to methods that are at risk for contamination with parasites and pathogens. Parasites, particularly intestinal helminths, may be a significant contributor to anemia in pregnancy in Uganda where intestinal worms was the $3^{rd}$ most common primary care diagnosis in 2014 [21]. Helminths can cause intestinal blood loss, contributing to iron deficiency anemia during pregnancy, which contributes to vascular consequences of the fetus, including growth restriction, as well as overall maternal mortality [22]. Additionally, opportunities to strengthen the collaboration between TBAs and the government midwives exist. The discord between what the midwives recommend during a pregnancy and what the TBAs are prescribing suggests that improved recommendations of safe and efficacious herbal medications free of parasites is warranted.

An important study finding is that women in Lweza desired more thorough education regarding pregnancy and child preparation. The WHO recommends having a standardized education program that teaches about healthy eating and nutrition, exercise during pregnancy, and non-pharmacologic options for symptoms during pregnancy [23]. Opportunities exist to address community-specific wants, such as family planning, HIV, complications of pregnancy, and depression education. A 2017 meta-analysis of women's groups who practiced participatory learning and action during the perinatal period, as recommended by the WHO, demonstrated the opportunity to adjust behaviors that are along the pathway to neonatal mortality [11]. Implementing the WHO antenatal care education delivery model may help address educational needs and improve maternal-neonatal health in Lweza.

## Conclusions

This study documents ways antenatal care and pregnancy health education is utilized in Lweza Village, Uganda. By gathering community member and community leader perceptions on

antenatal education and care, this study revealed opportunities for continued research and efforts, such as postpartum depression and herbal medicine, that might improve the care received by pregnant women in Lweza.

## Supporting information

**S1 File. Deidentified data.** This is the deidentified data from this study.
(XLSX)

**S2 File. Household questionnaire.** This is the questionnaire used with the 100 village women.
(DOCX)

**S3 File. Focus group discussion guide.** This guide was used to conduct the focus group discussion with both the teachers and the Village Health Project- Uganda groups.
(DOCX)

**S4 File. Key informant: Traditional birth attendant guide.** This interview guide was used to interview the TBA.
(DOCX)

**S5 File. Key informant: Village health team member guide.** This guide was used to interview a Village Health Team member.
(DOCX)

**S6 File. Key informant: Midwives guide.** This guide was used to interview the midwives.
(DOCX)

## Acknowledgments

We want to thank Mukasa Nsimbe Ronald for the project coordination and translating, and Nabatanzi Harriet who helped get respondents ready for interviews.

## Author Contributions

**Conceptualization:** Mackenzie E. Delzer, Ryan M. McAdams.

**Data curation:** Mackenzie E. Delzer.

**Investigation:** Mackenzie E. Delzer.

**Methodology:** Mackenzie E. Delzer, Ryan M. McAdams.

**Project administration:** Ryan M. McAdams.

**Resources:** Ryan M. McAdams.

**Supervision:** Anthony Kkonde, Ryan M. McAdams.

**Writing – original draft:** Mackenzie E. Delzer, Ryan M. McAdams.

**Writing – review & editing:** Mackenzie E. Delzer, Anthony Kkonde, Ryan M. McAdams.

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
