## [Decision Letter · Decision Letter 0]

16 Nov 2020

PONE-D-20-05007

Viewpoints of Pregnant Mothers and Community Health Workers on Antenatal Care in Lweza Village, Uganda

PLOS ONE

Dear Dr. Carlson,

Thank you for submitting your manuscript to PLOS ONE. After careful consideration, we feel that it has merit but does not fully meet PLOS ONE’s publication criteria as it currently stands. Therefore, we invite you to submit a revised version of the manuscript that addresses the points raised during the review process.

Two experts in the field handled your manuscript, and we are very thankful for their time and contributions. Although some interest was found in your study, several questions and comments arose. Please address ALL of the reviewers' remarks in your revised manuscript.

We look forward to receiving your revised manuscript.

Kind regards,

Frank T. Spradley

Academic Editor

PLOS ONE

"Research support was provided by an award from the University of Wisconsin School of Medicine and Public Health and the Herman and Gwendolyn Shapiro Foundation."

"None."

4. Please amend your current ethics statement to address the following concerns: Please explain why written consent was not obtained, how you recorded/documented participant consent, and if the ethics committees/IRBs approved this consent procedure.

5. During our internal checks, the in-house editorial staff noted that you conducted research in another country. Please check the relevant national regulations and laws applying to foreign researchers and state whether you obtained the required permits and approvals. Please address this in your ethics statement in both the manuscript and submission information.

6. Please include a copy of the interview guide used in the study, in both the original language and English, as Supporting Information, or include a citation if it has been published previously.

7. Please include additional information regarding the survey or questionnaire used in the study and ensure that you have provided sufficient details that others could replicate the analyses. For instance, if you developed a questionnaire as part of this study and it is not under a copyright more restrictive than CC-BY, please include a copy, in both the original language and English, as Supporting Information.  If the original language is written in non-Latin characters, for example Amharic, Chinese, or Korean, please use a file format that ensures these characters are visible.

8. Please state whether you validated the questionnaire prior to testing on study participants. Please provide details regarding the validation group within the methods section.

Reviewers' comments:

Reviewer's Responses to Questions

**Comments to the Author**

1. Is the manuscript technically sound, and do the data support the conclusions?

Reviewer #1: Yes

Reviewer #2: Yes

2. Has the statistical analysis been performed appropriately and rigorously? 

Reviewer #1: Yes

Reviewer #2: Yes

3. Have the authors made all data underlying the findings in their manuscript fully available?

Reviewer #1: No

Reviewer #2: Yes

4. Is the manuscript presented in an intelligible fashion and written in standard English?

Reviewer #1: Yes

Reviewer #2: Yes

5. Review Comments to the Author

Reviewer #1: Thank you for doing this study. It is a very interesting topic.

My recommendations are:

Text

• VHT is mentioned many times throughout the paper, however there is no description of what a VTH is or what the acronym means

• Line 381 – PDD should be PPD

• Line 147 – I would recommended expanding a little on your process of analysis Reference list:

• Please check that they meet the criteria for the journal – some information appears to be missing

• Reference 3 – can you please update the link so that the reader can find the article with this information easier

• Reference 5 – the link did not come up with anything, nor searching the WHO website, please update the reference

Figure 1 &2

• Can there be sharper pictures – they are blurry

Reviewer #2: Thank you for this interesting manuscript. I still have several questions about the study after reading the manuscript and some suggestions for clarification.

1. In the abstract, please identify the language used for data collection. It's located too far into the paper.

2. Explain the level of education of TBA and traditional healers if any. Do traditional healers get any training from other healers e.g. apprenticeship?

3. What is the population of the target village? The method of study population reads like more than 1 village was used for data collection since when the end of the village was reached, the team chose a different direction to continue data collection. These details should be clarified. Also explain what a boda boda is. What is the Child Care and Youth Empowerment Foundation? Your research funder? another organization helping in Uganda?

4. Did the interviewed village women receive any recompense for their time? This seems unfair if the key informants were reimbursed for time and the focus group participants received something of value.

5. For the common themes, provide information as to which themes emerged from which group (key informants, villagers, teachers, etc).

6. Identify what the literacy level is of the village women. Are pamphlets available about ANC that they could take home and read later for the topics they missed? Is any written information available or is it all verbal?

7. Explore the connection between the emumbwa and pica. I have known women in US who were told by older women to eat clay to make the baby strong.

8. Include the level of breastfeeding in the village. Is it universal or is formula also promoted by TBA/midwives/VHT?

6. PLOS authors have the option to publish the peer review history of their article (what does this mean?). If published, this will include your full peer review and any attached files.

Reviewer #1: No

Reviewer #2: **Yes: **Elizabeth Reifsnider

---

## [Author Response · Author response to Decision Letter 0]

31 Dec 2020

Reviewer #1: 

VHT is mentioned many times throughout the paper, however there is no description of what a VHT is or what the acronym means.

-The acronym is explained in line 208

Line 381- PDD should be PPD

-Updated

Line 147- I would recommend expanding a little on your process of analysis.

Response: Thank you for the suggestions. Village Health Team has been added to line 202 with the acronym. We corrected the PPD typo in line 381. We have revised the Methods to include our process of analysis. Qualitative data was transcribed and analyzed by the authors using thematic analysis, similar to the steps recommended by Braun and Clarke (cited in the manuscript). To become familiar with the data, the first author (M.D.) read through all the transcripts and field notes recorded from the standardized questionnaire, in-depth interviews and focus group discussions multiple times. Data were coded, manually analyzed, and grouped by assessing for relationships between them to form sub-themes. Based on this process, the key themes examined included navigating the available healthcare systems, pregnancy health education, herbal medicine, common community concerns, family dynamics, and family planning.

Reference list: Please check that they meet the criteria for the journal—some information appears to be missing.

Reference 3- can you please update the link so that the reader can find the article with this information easier

Reference 5- the link did not come up with anything, nor searching the WHO website. Please update the reference.

Response: The reference list has been manually updated to include all missing information.

Response: Reference 3 is now reference 4, and the link is updated: 

http://hdr.undp.org/en/countries/profiles/UGA

Reference 5 is now reference 6. The link was updated:

https://www.who.int/publications/i/item/9789241549912

Figure 1 & 2: Can there be sharper pictures—they are blurry.

Response: Thank you for letting us know. Both images have been revised. The dpi for Fig 1 is now 500 and for Fig 2 it is 300.

Reviewer #2:

1. In the abstract, please identify the language used for data collection. It’s located too far into the paper.

Response: Data collection was done in English with a Luganda translator. This information has been updated in the Abstract.

2. Explain the level of education of TBA and traditional healers if any. Do traditional healers get any training from other healers e.g. apprenticeship?

Response: TBAs are typically trained by previous generations of healers with knowledge of ailments and treatments being passed down from their mothers or family members. It is common for practices to differ between healers from different families or in different locations. This information has been added to the manuscript.

3. What is the population of the target village? The method of study population reads like more than 1 village was used for data collection since when the end of the village was reached, the team chose a different direction to continue data collection. These details shuld be clarified. Also explain what a boda boda is. What is the Child Care and Youth Empowerment Foundation? Your research funder? Another organization helping in Uganda?

Response: The population is roughly about 7,000 people. When we reached the end of Lweza Village, we randomly selected a new direction that would have us work back through Lweza instead of into a new village. A boda boda is a motorcycle taxi. The Child Care and Youth Empowerment Foundation is a local organization that was established to provide a gathering place for young girls and teenage mothers to share experiences, learn skills, and provide support to one another. All of these updates have been included in the revised manuscript.

4. Did the interviewed village women receive any recompense for their time? This seems unfair if the key informants were reimbursed for time and the focus group participants received something of value.

Response: Thank you for this question and comment. The village women did not receive any compensation for the interview. This was because the time commitment to complete the interview was much less. The key informant interviews and focus group discussions required more than an hour of time as well as some travel to get to a common gathering place. The household interviews with village women were much quicker, roughly 15-20 minutes, and the research team met women at their doors.

5. For the common themes, provide information as to which themes emerged from which group (key informants, villagers, teachers, etc)

Response: Thank you for this suggestion. We have added the groups and the themes that emerged from these groups to Table 2.

6. Identify what the literacy level is of the village women. Are pamphlets available about ANC that they could take home and read later for the topics they missed? Is any written information available or is it all verbal?

Response: The literacy level of the women interviewed was not obtained, however a general idea of literacy level may be inferred by highest level of education which was obtained. The education sessions at the government health facility in the morning were conducted primarily verbally. It is uncommon to give women reading materials at their appointments; however, at the antenatal clinic, there are a few posters displayed with some educational materials. This information has been added to the Results section.

7. Explore the connection between the emumbwa and pica. I have known women in US who were told by older women to eat clay to make the baby strong.

Response: Thank you for this suggestion. The following was added to the herbal medicine paragraph in the discussion section: 

“Emumbwa, a supplement commonly used during pregnancy, is made from soil and clay from the ground combined with a mixture of various local herbs. Consumption of this mixture could be considered a form of pica. Pica is the craving and consumption of nonnutritive substances, including soil, which is a global practice. (18) Pica is prevalent in Sub-Saharan Africa, especially in pregnant women, and while the exact etiology is unknown, pica may help satisfy micronutrient deficiencies and is considered a cultural norm. (18)”

8. Include the level of breastfeeding in the village. Is it universal or is formula also promoted by TBA/midwives/VHT?

Response: Based on observation, breastfeeding in the village seems to be universally promoted by women, midwives, VHTs, and TBAs. There is a cultural emphasis on breastfeeding, and access to formula is limited. In cases where breastmilk supplementation is warranted, like in HIV-positive mothers or where breastmilk production is lacking, it is commonplace to encourage supplementation with cow’s milk which may be easier to access and more affordable than formula. This information has been added to the Results section.

---

## [Decision Letter · Decision Letter 1]

18 Jan 2021

PONE-D-20-05007R1

Viewpoints of Pregnant Mothers and Community Health Workers on Antenatal Care in Lweza Village, Uganda

PLOS ONE

Dear Dr. Delzer,

Thank you for submitting your manuscript to PLOS ONE. After careful consideration, we feel that it has merit but does not fully meet PLOS ONE’s publication criteria as it currently stands. Therefore, we invite you to submit a revised version of the manuscript that addresses the points raised during the review process.

We look forward to receiving your revised manuscript.

Kind regards,

Frank T. Spradley

Academic Editor

PLOS ONE

Reviewers' comments:

Reviewer's Responses to Questions

**Comments to the Author**

1. If the authors have adequately addressed your comments raised in a previous round of review and you feel that this manuscript is now acceptable for publication, you may indicate that here to bypass the “Comments to the Author” section, enter your conflict of interest statement in the “Confidential to Editor” section, and submit your "Accept" recommendation.

Reviewer #1: All comments have been addressed

2. Is the manuscript technically sound, and do the data support the conclusions?

Reviewer #1: Yes

3. Has the statistical analysis been performed appropriately and rigorously? 

Reviewer #1: I Don't Know

4. Have the authors made all data underlying the findings in their manuscript fully available?

Reviewer #1: Yes

5. Is the manuscript presented in an intelligible fashion and written in standard English?

Reviewer #1: Yes

6. Review Comments to the Author

Reviewer #1: Thank you for addressing all my suggestions

I think it reads well however if there is going to be further review I recommend:

- Ensuring that all the acronyms are entered after the first full text is provided

- Consider reducing the amount of acronyms

- That your review to positioning of the additional informaton provided in lines 294-305 - would it fit better in another section of the paper?

7. PLOS authors have the option to publish the peer review history of their article (what does this mean?). If published, this will include your full peer review and any attached files.

Reviewer #1: No

---

## [Author Response · Author response to Decision Letter 1]

27 Jan 2021

Reviewer #1: Thank you for addressing all my suggestions

I think it reads well however if there is going to be further review I recommend:

- Ensuring that all the acronyms are entered after the first full text is provided

Response: Thank you. We have updated all acronyms so that they are entered after the first full text.

- Consider reducing the amount of acronyms

Response: Thank you. We reduced the amount of acronyms throughout the manuscript. The most common one used was for antenatal care (ANC), which appeared abundantly throughout the paper. Removal of “ANC” allows for easier reading. The acronyms “PPD” (postpartum depression) and “VHT” (Village Health Team member) were also eliminated to improve the flow of the paper.

- That your review to positioning of the additional information provided in lines 294-305 - would it fit better in another section of the paper?

Response: Thank you. The paragraph on literacy level was moved to line 281 where it fits with the demographic information about the respondents. The breastfeeding paragraph remained at line 336, where it fits in with the section about what women reported learning about.

---

## [Editor Report · Decision Letter 2]

29 Jan 2021

Viewpoints of Pregnant Mothers and Community Health Workers on Antenatal Care in Lweza Village, Uganda

PONE-D-20-05007R2

Dear Dr. Delzer,

We’re pleased to inform you that your manuscript has been judged scientifically suitable for publication and will be formally accepted for publication once it meets all outstanding technical requirements.

Kind regards,

Frank T. Spradley

Academic Editor

PLOS ONE

---

## [Editor Report · Acceptance letter]

3 Feb 2021

PONE-D-20-05007R2 

Viewpoints of Pregnant Mothers and Community Health Workers on Antenatal Care in Lweza Village, Uganda 

Dear Dr. Delzer:

I'm pleased to inform you that your manuscript has been deemed suitable for publication in PLOS ONE. Congratulations! Your manuscript is now with our production department. 

Kind regards, 

on behalf of

Dr. Frank T. Spradley 

Academic Editor

PLOS ONE